# Water Quality Evaluation and Pollution Source Apportionment of Surface Water in a Major City in Southeast China Using Multi-Statistical Analyses and Machine Learning Models

**DOI:** 10.3390/ijerph20010881

**Published:** 2023-01-03

**Authors:** Yu Zhou, Xinmin Wang, Weiying Li, Shuyun Zhou, Laizhu Jiang

**Affiliations:** 1College of Environmental Science and Engineering, Tongji University, Shanghai 200092, China; 2State Key Laboratory of Pollution Control and Resource Reuse, Tongji University, Shanghai 200092, China; 3Ministry of Education Key Laboratory of Yangtze River Water Environment, Tongji University, Shanghai 200092, China; 4Jiangsu Yinyang Stainless Steel Pipe Co., Ltd., Wuxi 214000, China; 5Fujian Qingtuo Special Steel Technology Research Co., Ltd., Fuzhou 350000, China

**Keywords:** water quality index (WQI), machine learning, parameter selection, positive matrix factorization (PMF), source apportionment

## Abstract

The comprehensive evaluation of water quality and identification of potential pollution sources has become a hot research topic. In this study, 14 water quality parameters at 4 water quality monitoring stations on the M River of a city in southeast China were measured monthly for 10 years (2011–2020). Multiple statistical methods, the water quality index (WQI) model, machine learning (ML), and positive matrix factorisation (PMF) models were used to assess the overall condition of the river, select crucial water quality parameters, and identify potential pollution sources. The average WQI values of the four sites ranged from 68.31 to 77.16, with a clear trend of deterioration from upstream to downstream. A random forest-based WQI model (WQI_RF_ model) was developed, and the results showed that Mn, Fe, faecal coliform, dissolved oxygen, and total nitrogen were selected as the top five important water quality parameters. Based on the results of the WQI_RF_ and PMF models, the contributions of potential pollution sources to the variation in the WQI values were quantitatively assessed and ranked. These findings prove the effectiveness of ML in evaluating water quality, and improve our understanding of surface water quality, thus providing support for the formulation of water quality management strategies.

## 1. Introduction

Surface water has historically been vital in providing water for human consumption, agriculture, and industrial requirements [1,2,3,4]. In recent decades, rapid urbanisation, industrialisation, and global population growth have led to the deterioration of surface water quality, which is a serious concern for the public and scientists [5,6]. According to a study conducted by the World Health Organization [7], at least 2 billion people worldwide use contaminated drinking water sources, 785 million people do not even have essential drinking water services, and 144 million rely on surface water.

As a water quality assessment method widely used for groundwater and surface water (especially rivers), the water quality index (WQI) method is playing an increasingly important role in water resource management [3,8,9,10]. Over the last several decades, various improvements have been made in the calculation of WQI values [11,12,13]. Compared with traditional water quality evaluation methods, the WQI method combines several environmental parameters, effectively transforming them into a single value reflecting the general water quality status, instead of comparing different evaluation results of various parameters [3].

To simplify and efficiently assess water quality, a WQI_min_ model based on a select number of representative parameters can quickly and accurately determine water quality and reduce analytical costs [14,15,16]. To determine the water quality parameters in the WQI_min_ model, previous studies mostly used linear regression methods based on the relationship between WQI values and various water quality parameters, and selected important indicators based on the performance of the WQI_min_ model on comprehensive evaluation values [3,10].

Machine learning (ML) models perform well in regression problems and have become very popular in recent years. In the field of environmental science, many scientists have used ML for water quality prediction. Chen et al., compared the water quality prediction performance of 10 ML models using big data from major rivers and lakes in China, identified two key water parameter sets (dissolved oxygen (DO), potassium permanganate index (COD_Mn_), and ammonia nitrogen (NH_3_-N); and COD_Mn_ and NH_3_-N), and proved the superiority of random forests (RFs) [17]. Lu and Ma used two hybrid models (extreme gradient boosting and RFs) to predict six water quality indicators (water temperature, DO, pH, specific conductance, turbidity, and fluorescent dissolved organic matter) and compared the performance of each model with those of four conventional models [18]. The results showed that the RF model had a higher prediction stability. In the present study, an RF model was used for regression modelling of WQI values, and important water quality parameters were selected according to the feature importance of RFs [19,20,21]. Selected key water quality parameters were then applied to develop the RF-based WQI_RFmin_ model.

In addition to completing the water quality assessment and obtaining important water quality indicators, it is also necessary to explore the potential sources of water pollution. Receptor models, such as the absolute principal components score combined with multivariate linear regression (APCS–MLR) and positive matrix factorisation (PMF), have performed well in source apportionment studies [22]. The PMF approach is a multi-source analysis method for source identification and assignment that is specifically designed to process environmental data and manage the associated uncertainty and distribution [23]. The PMF method is particularly suitable for environmental data because it considers the analytical uncertainty typically associated with environmental sample measurements and renders all values and contributions in the solution to be positive, which may lead to more realistic results than other multivariate methods [24]. Previous studies [22,25] showed that PMF had a higher coefficient of determination (R^2^) of prediction and a smaller proportion of unidentified sources than the APCS–MLR model, which could provide a more physically plausible source apportionment and a more realistic representation of pollution. In the last two decades, PMF has been widely used in studies related to air pollution and the atmospheric environment. In recent years, PMF has been increasingly used to apportion pollution sources in water environments [26,27]. The PMF model can describe the contributions of pollution sources to various water quality parameters; however, each water quality parameter has a different importance in different areas of research. Previous studies have rarely examined the contribution of pollution sources to WQI values, which can comprehensively assess water quality. Although some pollution sources provided a higher pollution contribution rate to water quality parameters in some studies, these sources may not be the main factor influencing water quality changes, because the concentrations of water quality parameters affected by them were too low to influence water quality changes [5].

The M River is an important river flowing through the capital city (mainly urban areas) of a province in southeast China, providing a permanent source of water for approximately 14 million people [28]. Based on the above background, WQI calculations, RF model construction, and PMF analyses were performed using a dataset of 14 water quality parameters collected on a monthly basis over 10 years (2011–2020) from four monitoring stations on the M River. The objectives of this study are to (1) analyse the spatial and temporal water quality patterns of the M River, (2) assess the comprehensive water quality condition and identify key water quality parameters of the M River, and (3) explore the potential pollution sources in the watershed and their contributions to the variation in WQI values. The results of the water quality assessment, crucial water quality parameter selection, and pollution source apportionment will be valuable for the local authorities to control and manage the water quality of the M River and to better protect it from pollution through a fixed-point traceability approach.

## 2. Materials and Methods

### 2.1. Study Area

The M River is located in the 25–29° N latitude and 116–120° E longitude region, and flows eastward through the Taiwan Strait. The river provides important assistance to people’s daily lives, industry, and agriculture in the cities of southeast China [28]. As a subtropical mountain river, the M River basin has an average annual temperature of 16–20 °C, and total annual rainfall of 1500–2000 mm, which is higher than that of other plain-dominated rivers in China. In recent years, modern agriculture has developed rapidly. The overuse of chemical fertilisers and pesticides, and the reckless discharge of sewage have intensified river pollution. Meanwhile, the continuous industrialisation and urbanisation of the M River basin have led to an increase in illegal discharges of industrial wastewater and an increase in heavy metal pollution due to mining, urban construction, and the development of transportation. Inadequate management of municipal, industrial, and agricultural wastewater means that residents around the watershed are exposed to dangerous organic and inorganic contamination of their drinking water [7,10,29].

### 2.2. Data Preparation

The datasets were collected on a monthly basis from October 2011 to August 2020 at four monitoring stations on the M River (WWP, FWP, SWP, and CWP; Figure 1). Fourteen water quality parameters were monitored as follows: pH, water temperature (WT), DO, total nitrogen (TN), NH_3_-N, nitrate-nitrogen (NO_3_^−^-N), total phosphorus (TP), COD_Mn_, chloride (Cl^−^), sulfate ion (SO_4_^2−^), faecal coliform (*F. coli*), iron (Fe), manganese (Mn), and fluoride (F^−^). The analytical methods used for each parameter are listed in Table 1.

### 2.3. Water Quality Index

The calculations for the WQI in this study are based on Equation (1), which was refined and developed by Pesce and Wunderlin [16] as follows:(1)$$WQI=\frac{\sum_{i-1}^{n}\left(C_{i}P_{i}\right)}{\sum_{i-1}^{n}P_{i}}$$
where n is the total number of water quality parameters in the study; $C_{i}$ is the normalized value of the *i*-th parameter; and $P_{i}$ is the determined weight of the *i*-th parameter (the values of $P_{i}$ have been verified in previous studies and are listed in Appendix A).

The theory of the WQI model has been widely used and extensively discussed in previous studies [2,3,29]. The water quality status in this study was classified into five grades based on the WQI values (Table 2), which are in line with the actual water quality management standards in China [3].

### 2.4. Random Forests

Random forest regressors are widely applied in ML for classification and regression, which can deal with nonlinearities and interactions, but cannot be interpreted directly [4,20,30]. It is an ensemble model based on the generation of many decision trees and their assemblage to produce the final output. Each output from the decision tree is dependent on the values of a random vector sampled independently from the same distribution of all decision trees generated in the forest. The number of predictors used to find the best split at each node is randomly chosen from a subset of all predictors [21]. The output is calculated by taking the mean and aggregation of each individual component tree [21,31]. The RF model has been found to be reliable for evaluating the ranking of the most critical predictors in trophic status prediction [32] and for predicting groundwater arsenic contamination [33].

In the construction of the decision tree, the quality of the segmentation variables and segmentation points are generally measured by the impurity of the node after segmentation.
(2)$$G\left(x_{i},v_{ij}\right)=\frac{n_{left}}{N_{s}}H\left(X_{left}\right)+\frac{n_{right}}{N_{s}}H\left(X_{right}\right)$$
where $x_{i}$ is a segmentation variable; $v_{ij}$ is a segmentation value of the segmentation variable; $n_{left}$ is the number of training samples of the left child node; $n_{right}$ is the number of training samples of the right child node; $N_{s}$ is the number of training samples of the current node; $X_{left}$ is the set of training samples of left child nodes; $X_{right}$ is the set of training samples of the right child nodes; *H(X)* is the impurity function of the node (classification and regression generally use different impurity functions).

The mean square error (MSE) was selected by default as the impurity function of the RF regression models based on decision trees as follows:(3)$$G\left(x,v\right)=\frac{1}{N_{s}}\left[\underset{y_{i}\epsilon X_{left}}{\sum }{\left(y_{i}-{\bar{y}}_{left}\right)}^{2}+\underset{y_{j}\epsilon X_{right}}{\sum }{\left(y_{i}-{\bar{y}}_{right}\right)}^{2}\right]$$

The importance of a node is given by:(4)$$n_{k}=w_{k}\times G_{k}-w_{left}\times G_{left}-w_{right}\times G_{right}$$
where $w_{k}$ is the ratio of the number of training samples to the total number of training samples in node k; $w_{left}$ is the ratio of the number of training samples in the left child node of node k to the total number of training samples in node k; $w_{right}$ is the ratio of the number of training samples in the right child node of node k to the total number of training samples in node k; $G_{k}$ is the impurity of node k; $G_{left}$ is the impurity of the left child node of node k; and $G_{right}$ is the impurity of the right child node of node k.

After calculating the importance of each node, the importance of a certain feature can be obtained as follows:(5)$$f_{i}=\frac{\sum_{j\epsilon nodes\;split\;on\;feature\;i}n_{j}}{\sum_{k\epsilon all\;nodes}n_{k}}$$

To ensure that the importance of all features will add up to one, the importance of each feature must be normalised:(6)$$f_{ni}=\frac{f_{i}}{\sum_{j\epsilon all\;features}f_{j}}$$

In this study, the WQI_RFmin_ model based on the key parameters selected by the RF regression model was also developed. The RF in this study consisted of 500 trees and was applied to train the WQI_RF_ model with the values of water quality indicators as the feature input model and the corresponding WQI as the label (predicted value), which were built using the Scikit-learn v.0.23.1 package in Python 3.8.3. Metrics including R^2^, MSE, MAE, and MAPE were adopted to evaluate the performance of the regressor on the testing dataset.

### 2.5. Positive Matrix Factorisation

The PMF method is a multivariate statistical analysis tool [23], which is usually used to decompose the sample data matrix into two matrices: factor contributions and factor profiles, with the following formula:(7)$$X_{nm}=E_{nm}+\sum_{j=1}^{p}G_{np}\times F_{pm}$$
where $X_{nm}$ is the original matrix (*n × m*), representing n samples and m monitoring variables, which can be decomposed into two matrices *G_np_ (n × p)* and *F_pm_ (p × m)*; p is the number of calculated sources (extraction factor); G is the source contribution matrix; F is the source component spectral matrix (factor load); *E_nm_ (n × m)* is the residual matrix representing the difference between the analytical result and the measured value.

The results are constrained by a penalty function such that no sample can have a negative source contribution, and no species can have a negative concentration in any source profile. A detailed description of the PMF model is provided in Paatero and Tapper [23]. The researchers have explained the PMF model in detail, thus no more detailed description here. This study used the PMF 5.0 software recommended by the US EPA for data analysis.

### 2.6. Contribution of Potential Pollution Sources to the Variation in WQI Values

According to the principle of RFs described in the previous section, the WQI_RF_ model based on water quality parameters was developed to quantitatively calculate the feature importance of each water quality parameter. The PMF model can quantitatively evaluate the contribution of each source to water quality; however, the WQI_RF_ model has calculation errors; therefore, $\left(1-{\mathrm{MAPE}}_{{\mathrm{WQI}}_{\mathrm{RF}}}\right)$ should be added as the error correction factor for the contribution of potential pollution sources to the variation in WQI values, as follows:(8)$$p_{j}=\left(1-{\mathrm{MAPE}}_{{\mathrm{WQI}}_{\mathrm{RF}}}\right)\times \sum \left(f_{ni}\times c_{ji}\right)$$
where $p_{j}$ is the contribution of pollution source j to the comprehensive water quality evaluation based on WQI values; ${\mathrm{MAPE}}_{{\mathrm{WQI}}_{\mathrm{RF}}}$ is the mean absolute percentage error of the WQI_RF_ model developed by RFs; and $c_{ji}$ is the contribution of pollution source j to water quality parameter i.

## 3. Results

### 3.1. Analysis of Water Quality Characteristics Based on Individual Parameters

The descriptive statistics of the original data for the selected 14 water quality parameters are listed in Appendix A. For water quality comparison, the surface water quality standards of GB3838-2002 (State Environment Protection Bureau of China 2002a) are also included in Appendix A. The statistical analysis results of each water quality parameter from 2011 to 2020 showed that, excluding TN, Fe, Mn, and *F. coli*, most of the water quality parameters were better than the Class III water quality standards over the long term.

Water pH indicates an acidic or basic nature and is an important parameter for assessing the quality of drinking water and irrigation water. It has profound effects on water quality, affecting the solubility of metals, alkalinity, and water hardness. From the analysis results, the incoming water from the four monitoring stations in River M over the past 10 years was relatively weakly acidic. The pH values ranged from 6.47 to 7.6, with 64% of the samples having a pH less than 7. Although it is in line with the surface water environmental quality standard GB3838-2002 (6–9 pH), but as a drinking water intake point, it is not enough to meet the surface water standard, but also needs to meet the drinking water hygiene standard GB5749-2022 (6.5–8.5 pH), which could only be said to just satisfy. As we all know, long-term consumption of acidic or weakly acidic water not only leads to the potential risk of erosive tooth wear, but also leads to gradually acidic body fluids, increased blood viscosity and imbalance of the acid–base balance of the human body. Many studies have shown that a low pH of the water supply system has a strong corrosive effect on metal pipes, which can easily lead to ‘yellow water’ and pipe bursts.

The values of TP, SO_4_^2−^, NO_3_^−^, F^−^, COD_Mn_, Cl^−^, NH_3_-N, and DO were lower than the respective Class III standards. For TN, 75% of the samples exceeded the Class III standards. The highest TN concentration (4.76 mg/L) was 4-, 2-, and 1.5-times higher than the standards of classes III, IV, and V, respectively. We observed that the multi-year average concentration of TN was 1.54 mg/L, with 48% and 23% of all observed samples exceeding the Class IV and V surface water standards, respectively (Figure 2). When TN and TP in surface water exceed their respective standards, microorganisms proliferate, plankton grow vigorously, and waterbodies are prone to eutrophication. Considering that the TN concentration did not increase significantly from upstream to downstream, the background value of the upstream water was the main factor. The causes of pollution may have been due to agricultural fertiliser (NO_3_^−^-N fertiliser) pollution, residential sewage, and farming wastewater pollution.

In addition, Mn, Fe, and *F. coli* exceeded the Class III standards to different degrees. The *F. coli* concentrations in the downstream region were significantly higher than those in the upstream regions, implying that the urban section of the city is a source of faecal coliform pollution to the river, although the background value of upstream water cannot be ignored.

Trace metals may be present in natural surface water and groundwater, and can be sourced from either natural processes or human activities. Multiple metal ion analyses were performed, but only Fe and Mn concentrations were found to be above the analytical detection limits. The Fe and Mn concentrations of water samples ranged from 1.26 mg/L to 3.2 mg/L and 0.16 mg/L to 1.52 mg/L, respectively. The exceedance rates of the Fe and Mn concentrations at the WWP and FWP monitoring sites in the upper reach were significantly lower than those at the CWP and SWP monitoring sites in the lower reach. The Mn and Fe concentrations at the WWP and FWP sites were likely related to the interaction between water and ophiolitic rocks in the basin, whereby relatively high levels of Mn and Fe in the surrounding ore-bearing landmass could provide a source of these elements to the rivers flowing over this terrain. The relatively high Mn and Fe concentrations at the downstream sites of CWP and SWP were probably mainly influenced by anthropogenic contaminants.

The coefficient of variation (CV) is the most discriminating factor in the variability description; it can eliminate the influence caused by the difference of units and the mean value between two or more datasets. As shown in Appendix A, all parameters showed a CV value of between 3.5% and >100%, indicating great variability. Among them, Cl^−^ and *F. coli* had the largest variabilities, indicating that these water quality parameters were extremely unevenly distributed throughout the basin and were affected by external sources of pollution. In addition, most analysed parameters in water samples presented spatiotemporal variabilities, whereby the concentrations of Mn, Fe, and *F. coli* in the lower reach were significantly higher than those in the upper reach (Figure 2).

### 3.2. Water Quality Assessment Based on the WQI

To calculate the WQI values at each sampling point, the weight values were determined for each water quality parameter according to their relative importance in terms of the overall drinking water quality (Appendix A). A weight of 3 was assigned to the trace metals, which can have major effects on water quality, especially for drinking purposes [15]. The accumulation of trace metals in water indicates both natural or anthropogenic sources, and may affect human health at high levels. The parameters of COD_Mn_, NH_4_-N, and *F. coli* were also each assigned a weight of 3 by taking into consideration their importance in water quality [10,14]. The exceedance of these indicators could lead to the presence of excessive organic pollutants in surface water [15], causing lasting toxic effects on aquatic organisms, and compromising drinking water safety for humans. The lower weights of 1 and 2 were assigned to WT, pH, TN, NO_3_-N, TP, Cl^−^, SO_4_^2−^, and F^−^ because of their low importance in water quality [3,10]. Then, the relative weights ($P_{i}$) were computed for each parameter. The WQI values were calculated using Equation (1), and the water quality types were determined for each sampling point (Appendix A).

The WQI results showed the spatial profiles and annual patterns of the variations in surface water quality (Figure 3). A violin plot is a collection of box-line and density plots, which can be used to show the percentile points of the data by thinking in terms of box lines, and a density plot to show the ‘contour’ effect of the data distribution, where the larger the ‘contour’ is, the more concentrated the data is. Based on the WQI scores, 58.2% of water samples were rated as ‘good’, with an average WQI value of 72.1, while the remaining water samples were rated as ‘moderate’.

Regarding the spatial variation in the calculated WQI values, the water quality exhibited a clear trend of deterioration from upstream to downstream. The mean WQI values at the FWP (upstream), WWP (upstream), SWP (downstream), and CWP (downstream) sites were 77.2, 74.1, 71.2, and 68.3, respectively. Overall, 86.4%, 76.5%, 51.2%, and 34.5% of water samples from the FWP, WWP, SWP, and CWP sites were rated as ‘good’, respectively. From the above analysis, Fe, Mn, and *F. coli* increased from upstream to downstream. As these water quality parameters accounted for high weightings in the calculation of the WQI, they were largely responsible for the decline in the WQI.

The annual changes in WQI values suggested that the median and interquartile range of WQI values shifted upward during the study period, and the wide part of the distribution density also shifted upward, indicating that the water quality was continuously improved with time. During 2011–2015, 54.2% of water samples were rated as ‘moderate’. In 2015, only 27.8% of water samples were rated as ‘good’. However, 70% of WQI values exceeded 70 (i.e., ‘good’) after 2016. The water quality in 2020 was the best, and the average WQI was 78.5, with 87.5% of water samples being rated as ‘good’.

### 3.3. Selection of Key Water Quality Parameters

The WQI_RF_ model was developed using RFs with all 14 water quality parameters (training data:testing data = 9:1), and the results showed that Mn made the most significant contribution to the WQI values (Figure 4). The parameters of Fe, *F. coli*, and DO were selected sequentially, and the R^2^ values of the models were considerably increased. Additionally, TN slightly enhanced the performance of the model. Hence, Mn, Fe, *F. coli*, DO, and TN were established as essential and critical parameters in the training of the WQI_RFmin_ model.

According to the constructed judgement of RFs on the importance of water quality parameters, two, three, four, and five parameters were selected to develop WQI_RFmin_ models using RFs. The performance of each WQI_RFmin_ model was based on a comprehensive evaluation of the R^2^, MSE, MAE, and MAPE values (Table 3, Figure 5), indicating that increases in the parameters could better explain the variation in the WQI. Among the WQI_RFmin_ models, the WQI_RFmin_ model comprising Mn, Fe, *F. coli*, DO, and TN had the best R^2^ (0.96), MSE (1.77), MAE (1.06), and MAPE (1.47%) values, indicating that it was the best WQI_RFmin_ model for the study area.

Based on the results of measured water parameters, water quality can be accurately assessed by some procedures; however, it is costly and time-consuming to measure all water parameters in all types of surface water because of the various analytical requirements. Therefore, it is more practical to measure key parameters indicative of water quality rather than completely following the guidelines of GB3838-2002 to understand water quality. Moreover, it is of great significance to predict water quality based on the selection of indicative fundamental water parameters. The five water quality parameters extracted by RFs in this study could determine the WQI with a very high accuracy.

### 3.4. Pollution Source Apportionment Using the PMF Model

According to a quantitative analysis of pollution sources based on PMF, five factors were determined for the surface water of the study area (Figure 6). F1 was characterised as microbial contamination because of the high percentage contribution of *F. coli* (87.4%), which could be attributed to sewage discharge, potentially from a leak due to a sewer system malfunction [5]. F2 was characterised by high weightings of TN (67.2%), F^−^ (61.3%), SO_4_^2−^ (81.6%), Cl^−^ (80.6%), and NO_3_^−^ (69.0%). A large amount of rural land is distributed in the upstream region of the M River. Considering that fertilisers might be transported with surface runoff and discharged into the river, frequent agricultural activities might have been the main cause of the high levels of nitrogen [25], and F2 could be attributed to non-point source agricultural pollution [26]. F3 was the main contributor of WT (53.6%), DO (58.5%), and COD_Mn_ (56.4%), as well as TP and TN; therefore, F3 may correspond to unexplainable variability, which may be the result of a combination of natural factors and urban domestic sewage [22]. F4 was characterised by a significant contribution of TP (73.3%), which is an important indicator of eutrophication; hence, F4 may represent nutrient pollution, which could include runoff pollution from urban areas [34]. The contribution rates of F5 were concentrated on Fe (79.3%) and Mn (93.7%), representing the impact of heavy metal pollution. The Fe and Mn concentrations in the M River increased significantly from upstream to downstream, indicating the external input of heavy metals in the study area, for example, from the local mining industry.

### 3.5. Contribution of Pollution Sources to Variation of WQI Value

The contributions of each potential pollution source to the variation in the WQI values were calculated (Table 4). Heavy metal pollution had the greatest impact on the WQI values, with a contribution of 53.18%, and the Fe and Mn concentrations increased significantly from the upper reach to the lower reach, which had a significant impact on the overall water quality. Therefore, close attention should be given to heavy metal pollution of the M River. The second largest contributor was microbial contamination (*F. coli*, 18.15%), which fluctuated widely in the M River and played a critical role in the WQI value. Non-point source agricultural pollution contributed significantly to many water quality parameters, but its contribution to the variation in the WQI values was only 9.64%. The concentrations of F^−^, SO_4_^2−^, Cl^−^, and NO_3_^−^ were generally stable. The TN concentration was relatively high for a long time and severely exceeded the Class III standard; however, its impact on the water quality evaluation was not significant. The contribution of nutrient contamination was 6.73%, which was primarily due to TP; however, TP was of a relatively good status for a long time and did not play a key role in the comprehensive evaluation of water quality. Unexplained variability contributed 10.95% to the variation in the WQI values, in which DO was a crucial water quality parameter for the WQI.

## 4. Discussion

### 4.1. Quantitative Assessment of the Impact of Pollution Sources on Water Quality

The WQI can comprehensively evaluate the status of water quality. For the trained WQI_RF_ model based on RFs, according to the analysis of the model’s feature importance, the proposed WQI_RFmin_ model in this study consisted of five key water quality parameters, that is, Mn, Fe, *F. coli*, DO, and TN, and exhibited a very good performance for water quality evaluations. The selected parameters of the WQI_RFmin_ model should be able to comprehensively explain the overall variations and characteristics of water quality and should be conducive for efficiently evaluating water quality with relatively lower measurement costs [3]. Five potential pollution sources were obtained using the PMF method. Because the RF model could assess the importance of each parameter in the model, the feature importance of each water quality parameter in the WQI_RF_ could be calculated. The contribution of each potential pollution source to the variation in the WQI values was quantitatively assessed by multiplying the feature importance of each water quality indicator by the contribution of the source to each water quality indicator in the PMF model and then accumulating them.

Previous studies have used the WQI to assess surface water quality in many areas [2,3,8,9,35], and many studies have also analysed potential pollution sources of surface water [36,37,38]. However, the determination of most pollution sources and their effects are usually based on the personal experience of the researcher and the qualitative judgement of the local survey information [26].

Few studies have quantitatively analysed the impact of pollution sources on the water quality assessment. Although some pollution sources provided a higher pollution contribution rate to water quality parameters in this study, the contribution of the pollution source to the WQI values was not enough to change the WQI values; this, the actual impact of these sources on the water quality assessment was not significant. Through the quantitative analysis of the relationship between pollution sources and the WQI values, it is possible to (i) obtain the pollution sources that have a substantial impact on water quality evaluation, (ii) clarify the focus of water pollution management, and (iii) provide relevant departments with a reasonable water resource protection strategy.

From the perspective of water quality evaluation, this study systematically analysed the water quality of the M River basin and obtained five important water quality indicators through the ML method. From the perspective of pollution source analysis, this study identified potential pollution sources and quantitatively analysed the impact of pollution sources on water quality evaluation.

The method used in this study identified the most important potential sources of pollution in terms of their effect on the WQI score. Nevertheless, the disadvantage of using the receptor PMF model to determine the potential sources of pollution in surface water is that the source of pollution to a waterbody cannot be clearly identified. If the potential sources of pollution can be identified by this method for targeted pollution control, and subsequent water samples can be collected and compared for water quality analysis, the results of present studies could be verified. Moreover, the important water quality indicators and water quality characteristics could also be analysed before and after pollution control.

### 4.2. Advantages and Innovation of RFs in the Construction of the WQI_min_ Model

In previous studies, scholars generally used the stepwise multiple linear regression method to develop the WQI_min_ models [3,10], which were evaluated based on R^2^, MSE, and percentage error (PE) values to select important water quality indicators. Compared with previous studies, the data distribution of WQI values in the present study was wide and the model was relatively difficult to construct. The WQI_min_ obtained with the above method did not perform well on the testing set, in which PE > 10% [10].

In recent years, ML has shown excellent performance in regression models, and has attracted increasing attention for use in academia and industry. The RF-based WQI_RFmin_ model in this study exhibited a better performance and yielded more stable results compared with the traditional stepwise multiple linear regression method (Appendix A). In recent years, some research has focused on combining ML with individual water quality indicators. Chen et al. used ML methods to classify surface water quality with only a few water quality parameters [17]. However, the national standards for surface water quality evaluation in China still use a single-indicator evaluation method. There are relatively few studies on the combination of ML and comprehensive water quality assessment. The use of RFs combined with the WQI method in this study is a novel attempt to use ML for water quality assessment. Given the rapid development of artificial intelligence and big data, ML and deep learning can be combined with water quality assessment, water quality warning systems, and other related water quality research in the future.

## 5. Conclusions

The main conclusions are as follows: (1) The main water quality parameters of the M River that exceeded the Class III standards were TN, *F. coli*, Fe, and Mn. The WQI results indicated that the water quality of the M River was ‘good’ overall, with an overall average WQI value of 72.11. The average WQI values of the four monitoring stations ranged from 68.31 to 77.16, and there was a clear trend of deterioration from upstream to downstream. (2) The feature importance of each water quality parameter in the WQI_RF_ model was quantitatively assessed, and five parameters (Mn, Fe, *F. coli*, DO, and TN) were selected as key water quality parameters for establishing the WQI_RFmin_ model, which had good accuracy (R^2^ = 0.96). (3) The PMF method was applied to identify five pollution sources and to apportion their contributions to each water quality parameter. (4) Quantitative assessment of the impact of pollution sources on water quality showed that pollutions sources were ranked as: heavy metal pollution (53.18%) > microbial contamination (18.15%) > non-point source agricultural (9.64%) > nutrient contamination (6.73%), while the unexplained variability accounted for 10.95% of the total.

The methods used in this study to analyse the water quality of the M River could reduce the measurement cost of water quality assessment and effectively improve the measurement efficiency. In addition, the findings provide support for formulating water quality management strategies. The methods of selecting key water quality parameters and of assessing the quantitative contributions of pollution sources to the variation in the WQI values could be practically applied to other surface waters to greatly improve our understanding of the overall water quality condition. Additional studies will be required to assess precisely the unidentified sources of pollution and variation of further water quality parameters that were not analyzed in this study.

However, water pollution is a complex process, and more factors will affect the migration and transformation of pollutants. Therefore, we should continue to improve the research methods and technical means, and explore the methods and theories of traceability of exceeded pollutants at both qualitative and quantitative levels. It is necessary to verify and analyze the existing results, optimize the sampling scheme, and establish a model of the relationship between environmental variables and water pollutants. This will be a major direction for future development.

## Figures and Tables

**Figure 1 ijerph-20-00881-f001:**
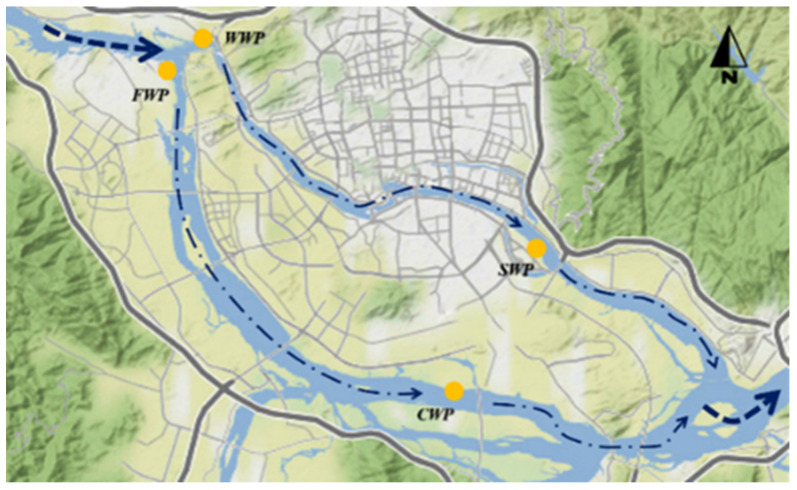
Locations of the water quality monitoring stations in the study area in southeast China.

**Figure 2 ijerph-20-00881-f002:**
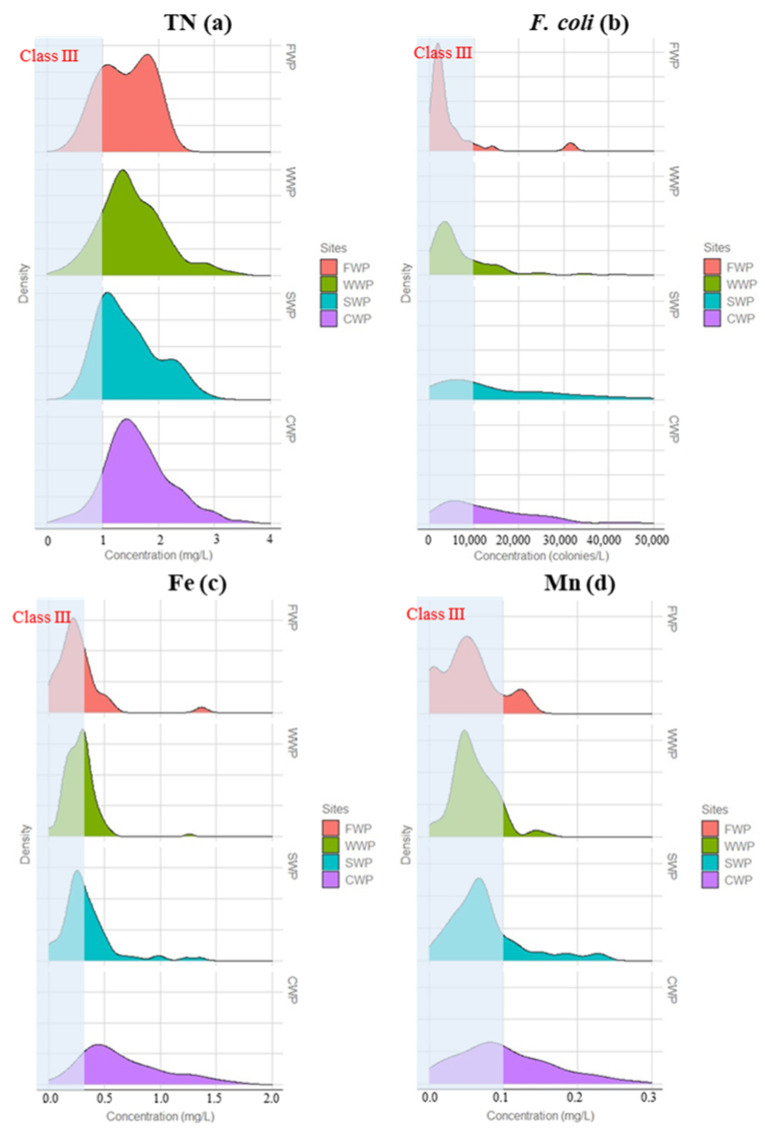
Density distributions of (**a**) TN, (**b**) *F. coli*, (**c**) Fe, and (**d**) Mn concentrations.

**Figure 3 ijerph-20-00881-f003:**
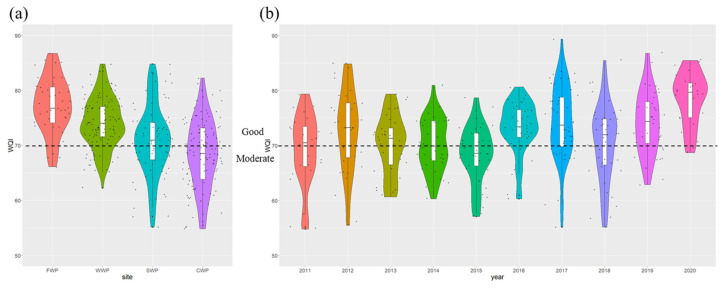
Spatial (**a**) and annual (**b**) variations of the WQI during 2011–2020.

**Figure 4 ijerph-20-00881-f004:**
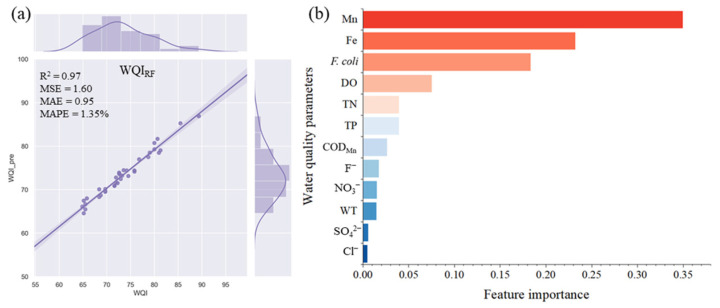
WQI_RF_ model results. (**a**) The predicted results on testing data and (**b**) the feature importance of key water quality parameters.

**Figure 5 ijerph-20-00881-f005:**
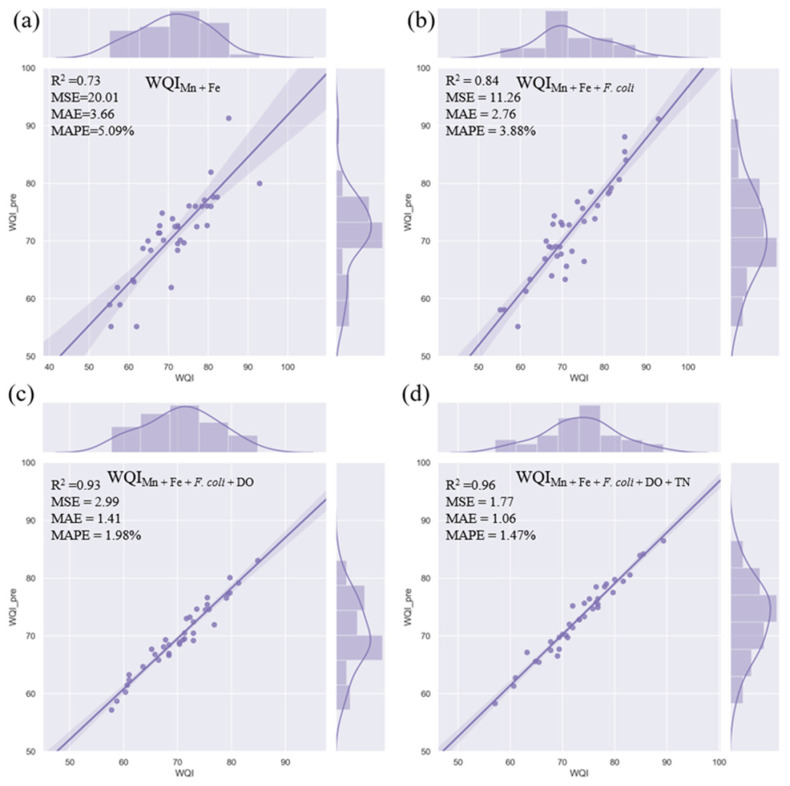
Comparison of the WQI_RFmin_ values based on different groups of parameters. (**a**) Mn + Fe, (**b**) Mn + Fe + *F. coli*, (**c**) Mn + Fe + *F. coli* + DO, (**d**) Mn + Fe + *F. coli* + DO + TN.

**Figure 6 ijerph-20-00881-f006:**
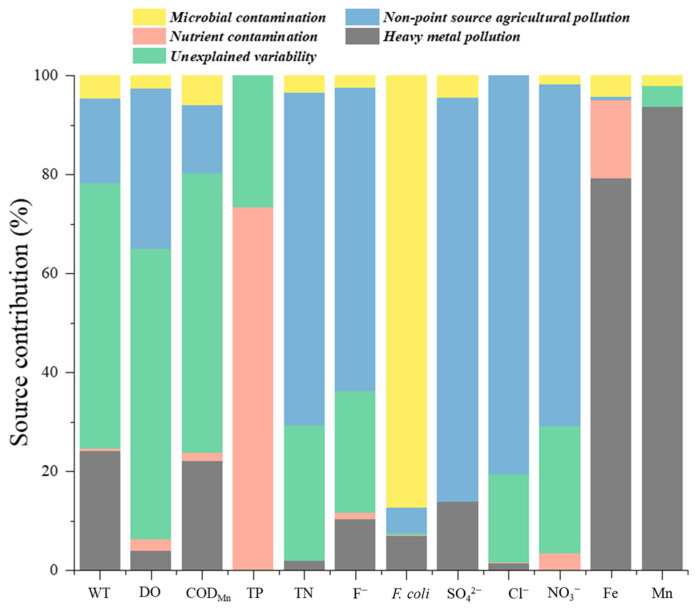
Contributions of pollution sources to the selected water quality variables.

**Table 1 ijerph-20-00881-t001:** Water quality parameters measured in this study and the relevant analytical methods.

Variables	Abbreviation	Units	Testing Base
pH	pH		GB6920-1986
Water temperature	WT	°C	GB/T13195-1991
Dissolved oxygen	DO	mg/L	HJ506-2009
Total nitrogen	TN	mg/L	HJ636-2012
Ammonia	NH_3_-N	mg/L	HJ665-2013
Nitrate	NO_3_-N	mg/L	HJ/T84-2001
Total phosphorus	TP	mg/L	GB/T11893-1989
Permanganate index	COD_Mn_	mg/L	GB 11892-1989
Chloride	Cl^−^	mg/L	HJ/T84-2001
Sulphate	SO_4_^2−^	mg/L	HJ/T84-2001
Iron	Fe	mg/L	HJ700-2014
Manganese	Mn	mg/L	HJ700-2014
Fecal coliform	*F. coli*	colonies/L	GB/T5750.12-2006
Fluoride	F^-^	mg/L	HJ/T84-2001

**Table 2 ijerph-20-00881-t002:** Water quality classification based on water quality index (WQI) values.

**WQI value**	91–100	71–90	51–70	26–50	0–25
**Water quality**	Excellent	Good	Moderate	Poor	Very poor

**Table 3 ijerph-20-00881-t003:** Parameter selection results of the WQI_RF_ models based on the training dataset.

Parameters	Feature Importance	R^2^	MSE	MAE	MAPE (%)
Mn	0.35	—	—	—	—
Mn + Fe	0.58	0.73	20.01	3.66	5.09
Mn + Fe + *F. coli*	0.76	0.84	11.26	2.76	3.88
Mn + Fe + *F. coli* + DO	0.84	0.93	2.99	1.41	1.98
Mn + Fe + *F. coli* + DO + TN	0.88	0.96	1.77	1.06	1.47
All water quality parameters	1	0.97	1.60	.0.95	1.35

**Table 4 ijerph-20-00881-t004:** Contribution of pollution sources to the variation in WQI values.

Pollution Sources	Microbial Contamination	Non-Point Source Agricultural Pollution	Unexplained Variability	Nutrient Contamination	Heavy Metal Pollution	Model Error
**Contribution (%)**	18.15	9.64	10.95	6.73	53.18	1.35

## Data Availability

Not applicable.

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
