# Peer review of "Water Quality Evaluation and Pollution Source Apportionment of Surface Water in a Major City in Southeast China Using Multi-Statistical Analyses and Machine Learning Models"

_ijerph, 2023, doi:10.3390/ijerph20010881_

Round 1

Reviewer 1 Report

I would prefer a space between the text and the reference: Paatero and Tap- 203 per[23].

This is recommended for all sentences with references  (Previous studies have used the WQI to assess surface water quality in many areas[2,3,8,9,35])  add the space before the brackets (Parentheses)

This for all sentences, it increases the readability

Table S2 (China, 2002a). references in the text are not consistent

Line 247 F. coli  Surely it is E coli, same for lines 248, 268, 272, 272, 283 as well as Figure 4B (321), line 329, Table 3, 341. Figure 5. Line 348, 459, 464.

I strongly recommend the authors do a find-and-replace to address this universal issue.

Author Response

Thank you for your effort and comments. Please see our response in the attachment.

Reviewer 2 Report

- line 231

Frame the statements properly. Like "From the analysis results, the surface water was slightly acidic at all four monitoring stations" needs to be reframed.

-line 233

Correct the grammatical errors. Like "pH of <7"

-line 239

Reframe the statement properly "The average TN concentration was 1.54 mg/L, with 48% and 23% of samples exceeding the standards of classes IV and V "

-Abbreviations have been used at many places in the text. Excessive and unnecessary use of abbreviations in the text makes the manuscript uninteresting for readers. Avoid too many abbreviations and every abbreviation must contain its full form when used first time.

-The manuscript needs to be checked properly for English errors and rectified.

-The methodology section must include elaborated information on how the machine learning tool was applied to water quality data (software/input values etc) while the discussion section should cover the output and its significance in deciding the quality of water.

-Lines 210-215

-If PMF is sufficient to find the pollution sources and the WQIRF model has calculation errors, then why WQIRF is additionally used? Please explain with reference to the following statement "The PMF model can quantitatively evaluate the contribution of each source to water quality; however, the WQIRF model has calculation errors; therefore, ...()...should be added as the error correction factor for the contribution of potential pollution sources to the variation in WQI values, as follows..()..."

-Write the full forms of MSE, MAE, and MAPE before using them in text repeatedly

-lines 229-235

Discussion of pH results with reference to the China River Water Quality standards (6-9 pH) must be added in the discussion section

-Provide the elaborated description of figure 3 in a few lines

Line 451-456

In the conclusion section, the text in lines 451-456, is just the repetition of methodology, and need not repeat here. "In this study.....values were quantitatively evaluated". 

The conclusion section must cover the concrete findings of the overall study and not the mere repetition of results and discussion.

Author Response

(The authors gave the same response as above.)

Reviewer 3 Report

The work deals with an important issue - the impact of a large urban agglomeration on shaping the quality of water in the river. The subject matter of the work is in line with research trends both in Central Europe and Scandinavia (e.g. Sweden).j
  In my opinion, the methodology of the work is correct, and the research results are credible and consistent with the results of research conducted in, among others, large European urban agglomerations. At the same time, I state that the article brings and extends new values and contribution to world science on the important issue of assessing the impact of urban agglomerations on the environment.

Author Response

(The authors gave the same response as above.)

Reviewer 4 Report

The paper describes a case study for testing a water quality index approach for assessing water quality in a river in China. The river is called 'M'ahd the city where the study takes place is not named, but from the information given, it can be easily seen that the study is done in Fouzhou.

The study shows not much novelty and it is not very clear what research question is actually addressed. The data used are very basic water quality parameters and don't include information on emerging contaminants etc. The study does not make a connection between the WQI and the potential impact of water quality on ecosystems, ecotoxicity or for instance production or drinking water.

The model also tries to do pollutions source apportionment, but does not go beyond very general pollution sources such as 'sewage discharge', 'agriculture' or 'mining', without identifying the real sources.

Author Response

(The authors gave the same response as above.)
